# Social Media and COVID-19—Perceptions and Public Deceptions of Ivermectin, Colchicine and Hydroxychloroquine: Lessons for Future Pandemics

**DOI:** 10.3390/antibiotics11040445

**Published:** 2022-03-25

**Authors:** Natalie Schellack, Morné Strydom, Michael S. Pepper, Candice L. Herd, Candice Laverne Hendricks, Elmien Bronkhorst, Johanna C. Meyer, Neelaveni Padayachee, Varsha Bangalee, Ilse Truter, Andrea Antonio Ellero, Thulisa Myaka, Elysha Naidoo, Brian Godman

**Affiliations:** 1Department of Pharmacology, Faculty of Health Sciences, University of Pretoria, Pretoria 0084, South Africa; morne.strydom@up.ac.za (M.S.); andrea.ellero@up.ac.za (A.A.E.); u17079421@tuks.co.za (T.M.); u17083932@tuks.co.za (E.N.); 2Institute for Cellular and Molecular Medicine, Department of Immunology and SAMRC Extramural Unit for Stem Cell Research and Therapy, Faculty of Health Sciences, University of Pretoria, Pretoria 0084, South Africa; michael.pepper@up.ac.za (M.S.P.); clherd@tuks.co.za (C.L.H.); candice_hendricks@outlook.com (C.L.H.); 3School of Pharmacy, Sefako Makgatho Health Sciences University, Pretoria 0084, South Africa; elmien.bronkhorst@smu.ac.za (E.B.); hannelie.meyer@smu.ac.za (J.C.M.); 4Department of Pharmacy and Pharmacology, Faculty of Health Sciences, School of Therapeutic Sciences, University of Witwatersrand, Johannesburg 2050, South Africa; neelaveni.padayachee@wits.ac.za; 5Discipline of Pharmaceutical Sciences, College of Health Sciences, University of KwaZulu-Natal, Durban 4041, South Africa; bangalee@ukzn.ac.za; 6Drug Utilization Research Unit (DURU), Department of Pharmacy, Nelson Mandela University, Port Elizabeth 6031, South Africa; ilse.truter@mandela.ac.za; 7Centre for Neuroendocrinology (CNE), Department of Immunology, University of Pretoria, Pretoria 0084, South Africa; 8Strathclyde Institute of Pharmacy and Biomedical Sciences, University of Strathclyde, Glasgow G4 0RE, UK; 9Centre of Medical and Bio-Allied Health Sciences Research, Ajman University, Ajman P.O. Box 346, United Arab Emirates

**Keywords:** social media, re-purposed medicines, hydroxychloroquine, ivermectin, colchicine, South Africa, sentiment analysis, utilization, clinical trials

## Abstract

The capacity for social media to influence the utilization of re-purposed medicines to manage COVID-19, despite limited availability of safety and efficacy data, is a cause for concern within health care systems. This study sought to ascertain links between social media reports and utilization for three re-purposed medicines: hydroxychloroquine (HCQ), ivermectin and colchicine. A combined retrospective analysis of social media posts for these three re-purposed medicines was undertaken, along with utilization and clinical trials data, in South Africa, between January 2020 and June 2021. In total, 77,257 posts were collected across key social media platforms, of which 6884 were relevant. Ivermectin had the highest number of posts (55%) followed by HCQ (44%). The spike in ivermectin use was closely correlated to social media posts. Similarly, regarding chloroquine (as HCQ is not available in South Africa), social media interest was enhanced by local politicians. Sentiment analysis revealed that posts regarding the effectiveness of these repurposed medicines were positive. This was different for colchicine, which contributed only a small number of mentions (1%). Of concern is that the majority of reporters in social media (85%) were unidentifiable. This study provides evidence of social media as a driver of re-purposed medicines. Healthcare professionals have a key role in providing evidence-based advice especially with unidentifiable posts.

## 1. Introduction

Human beings crave social interaction, which has been severely curtailed in response to the COVID-19 pandemic across Africa and other countries [1,2,3]. During the pandemic, the safest form of social contact was through social media, which plays an important role in the daily lives of many people [4]. This was because early control of COVID-19 centred around prevention. Measures included regular hand sanitising, wearing of masks, social-distancing and societal lockdown measures incorporating closure of borders and educational establishments, which were initially successful in a number of African and Asian countries [1,2,5,6,7,8,9,10,11]. 

Over the last decade, various social media platforms including Twitter, Facebook, YouTube, WhatsApp, LinkedIn and Instagram have become increasingly influential. These platforms are central to how people across the world communicate with each other and share information [12]. Major events that impact the public, such as the recent COVID-19 pandemic, prompt an increase in public desire to search for information surrounding the topic [13]. During the early stages of the global COVID-19 pandemic, people were unaware of ways to help themselves during the crisis. This confusion and inherent frustration resulted in the viral sharing of any available information relating to the pandemic through social media. This was illustrated by the surge in internet searches, or searches on social media, relating to pandemics, showing a correlation with the incidence of the disease [13]. 

However, social media usage during a global crisis such as the COVID-19 pandemic has both advantages and disadvantages. Advantages include the rapid distribution of educational information about the disease such as preventative measures and treatment possibilities [14,15]. Disadvantages include the possibility that the information spread may not be current, not factual, not subjected to peer review or false [13,16,17]. Social media may also create stress, uncertainty, fear and unnecessary panic among the public due to exaggerated information or fake news aimed at misleading people as seen in the current pandemic [13,18]. Many people also made use of self-medication or herbal medicines during the recent COVID-19 pandemic in the absence of effective medicines, certainly initially, guided by the influence of social media, which was a concern during the pandemic [19].

It is recognised that the development of new medicines typically takes many years from initiation of pertinent research to their approval as safe for human administration. Consequently, given the impetus to improve the management of patients with COVID-19 amidst growing morbidity and mortality rates, the repurposing of potential medicines became a priority although there are concerns [20,21,22]. The challenge in the case of COVID-19 was amplified by the uncertainty initially surrounding the disease, as well as the risk to benefit profile of potential treatments, which will differ among individuals and disease areas [23,24,25].

The urgent requirement to determine potential treatments for patients with COVID-19 initially resulted in a number of poorly controlled clinical trials with unreliable safety and efficacy data, exacerbated by often limited patient numbers in the studies [20,26]. This was the case of hydroxychloroquine (HCQ), which was one of the first medicines to be considered for repurposing for treating patients with COVID-19 [27,28]. This led the public to believe that HCQ, or chloroquine when HCQ was unavailable, may be useful for routine treatment, leading to increased use and shortages resulting in increased prices [3,29,30,31] enhanced by social media activities [32]. However, the questionable scientific rigor of the initial studies [22,28,33,34] and the lack of patient benefit in robust studies [35,36,37], coupled with the potential for adverse drug reactions and death from HCQ, and also chloroquine [38,39,40], resulted in HCQ no longer being recommended in international and national guidelines [41,42]; similarly, this was also the case for lopinavir/ritonavir, remdesivir, and interferon regimens [42,43,44,45].

Colchicine and ivermectin are other examples of repurposed medicines that have attracted considerable social media coverage. Colchicine is a drug approved for the treatment of gout. However, it has demonstrated anti-inflammatory properties which could aid in the systemic relief of inflammation caused by SARS-CoV-2 infections [46]. Karatza et al. proposed that colchicine may prevent the phenomenon referred to as the ‘cytokine storm’ and may reduce any resulting complications thereof [47]. However, the narrow therapeutic index associated with colchicine carries the potential for serious adverse effects. There is also uncertainty regarding its effectiveness in reducing mortality in patients with COVID-19 [47,48,49].

Ivermectin is an anthelmintic medicine used to treat parasitic infections. It was originally targeted for the treatment of animals; however, it has now been shown to be safe in humans for the treatment of parasitic infections [50]. In addition to its anti-parasitic mode of action, ivermectin has been shown to inhibit α/β mediated nuclear import of viral particles by the inhibition of the interaction between integrase and importin [51]. This is an important mechanism, which subsequently reduces viral replication. Despite the reports in several current studies, along with a recent systematic review and meta-analysis published by Roman et al. showing that ivermectin is not a viable option to treat patients with COVID-19, its popularity on various social media platforms continues to rise with a proportional increase in utilisation [52,53,54].

To date, out of HCQ, colchicine and ivermectin, none have been approved for the treatment of COVID-19 in South Africa. Both the South African Health Products Regulatory Authority (SAHPRA) and the National Department of Health in South Africa have issued multiple statements and reviews, respectively, against the use of these three medicines to treat patients with COVID-19 [55,56,57]. However, this has not deterred the inappropriate use of these unproven medicines among the public for the prevention and treatment of COVID-19.

This is an issue as international and national regulatory agencies including the USA Food and Drug Administration (FDA), the UK Medicines and Healthcare Products Regulatory Agency (MHRA) and SAHPRA aim to protect the public by only approving medicines that demonstrate a favourable risk-to-benefit ratio. Regulatory agencies should serve to ensure that relevant and up-to-date information on medicines is available to all healthcare professionals (HCPs) and patients in a timely manner [1].

This is because both the general public and HCPs can be susceptible to biases when making health-related decisions [24], which can be exacerbated in times of crisis. This is a concern as misinformation during the current pandemic increased depression, fatigue, fear, and panic [18].

Recently some African countries considered the spread of misinformation a legal offence, prosecuting people and organisations for spreading it [2], with such activities likely to grow [58,59]. In the meantime, there is a need to actively research social media activities and their impact in more detail [4].

Consequently, the objective of this study was to analyse the impact of social media as a potential driver for the utilisation of repurposed medicines in the management of COVID-19 by focusing on the three most popular publicly repurposed medicines namely hydroxychloroquine, colchicine and ivermectin and the situation in South Africa. The findings should be useful in helping to guide the distribution of information to all key stakeholder groups.

This is especially important, given the potential for increasing morbidity, mortality and costs with the inappropriate use of repurposed medicines to manage patients with COVID-19.

## 2. Results

### 2.1. Summary of Posts Analysed

A total of 77,257 posts were collected across all social media platforms in South Africa, and after filtration and contextualisation methods were applied, 6884 were found to be relevant to the study objective. A total of 1070 unique authors/reporters were identified from the posts relevant to the objective, accounting for authors/reporters that may have made multiple posts regarding different topics. The major digital platforms hosting the majority of the accessible data included Twitter, news websites, forums, Tumblr and blogs. Table 1 summarises the number of posts analysed across all platforms.

### 2.2. Social Mentions of Repurposed Medicines 

Ivermectin had the highest number of social media mentions (55%), followed by HCQ (44%) and colchicine (1%) (Figure 1). Twitter was the platform with the most mentions of the three repurposed medicines, followed by news sites and forums. 

News sites, blogs and forums were extensively used to share articles on the experiences of the various medicines, political campaigns, and clinical research. These were extensively used by healthcare professionals (HCPs), news organisations and online users in South Africa for social media mentions (Figure 1). The themes of conversation specific to news platforms consisted of 39% treatment-related themes, 31% related to the approval of these medicines, 21% related to clinical trials and research, 5% the drug’s safety profile and 4% politically related.

### 2.3. Ivermectin

Overall, ivermectin was the most popular of the three repurposed medicines being monitored, contributing 55% of total social media posts analysed and totalling 42,255 posts. In this analysis, ivermectin discussions began in January 2021 and are ongoing. There were three dominant peaks in social media mentions involving ivermectin, in January 2021, April 2021 and July 2021 (Figure 2A(i–iii)). 

Social media conversations for ivermectin peaked in January 2021 following a tweet on Dr. Naseeba Kathrada’s letter to the President of South Africa seeking approval for its use in patients with COVID-19. People used blogs such as Netwerk24to share previous experiences with ivermectin and discus its effectiveness in the treatment of COVID-19. The prominent topics of discussion were the consistent lack of evidence-based data. The top shared articles for ivermectin focused on the struggle to get the medicine approved by SAHPRA. Sentiment analysis revealed that 80% of posts were positive and 20% were negative. Patients and caregivers strongly voiced their opinion regarding ivermectin and its effects. The general perception was that ivermectin is effective in providing quick relief to COVID-19 symptoms.

Ivermectin utilisation (Figure 2B, Appendix A) increased from 1090 total recorded units in 2019, to 1516 units in 2020 and 8100 units in 2021. Formulations monitored were subcutaneous injectable preparations indicated for use in animals only and oral tablets (Figure 2C,D). Utilisation data for oral tablets were only recorded in the first half of 2021 due to their previous unavailability within South Africa. Unit increases of over 740% and 530% were noted when comparing 2021 to 2019 and 2020, respectively.

This spike in moving annual total (MAT) units is closely correlated to the increases in popularity for ivermectin on social media. The persistence of this trend within formulations indicated for subcutaneous animal use is concerning and may be an indication of public self-prescribing and purchasing the medication themselves.

### 2.4. Chloroquine (Hydroxychloroquine)

As mentioned, HCQ is currently not approved for use in South Africa. Consequently, its metabolic precursor, chloroquine, which is approved for use in South Africa, was monitored.

Chloroquine demonstrated a compound annual growth rate (CAGR) of 26% and 33% for MAT units/scripts and Rand value, respectively, from 2019 to 2020 indicating a likely price increase occurred during the timeframe accounting for the disparity (Appendix A). Price increases have also been seen in other African countries following the pandemic [11].

Social media trends (Figure 3A(i–iii)) were indicative of high public interest in the compound evident by the high number of total mentions and sharp peaks. Increases in chloroquine utilisation also closely followed the increases in social media mentions (Figure 3B).

Among the four chloroquine formulations, chloroquine 200 mg capsules represented the highest market share (in the 90th percentile) for all the specified years as compared to the other formulations (Figure 3B). This is likely due to the advantages that capsule formulations offer over others, including the ease of administration, accurate dosing without measuring and avoidance of poor taste associated with syrups.

Overall, 44% of total social media posts analysed were related to HCQ. HCQ was discussed mainly at the start of the pandemic from March 2020 until August 2020 when discussions surrounding HCQ started to decrease. In total, there were 34,464 posts related to HCQ over the whole timeline, from January 2020 to June 2021. There are three major peaks visible on the trendline, in March 2020, May 2020 and July 2020. The hashtag ‘#hydroxychloroquine’ was utilised by online users while sharing tweets on the side-effects of HCQ and any ongoing clinical trials for its use to treat COVID-19. 

On social media, HCQ was discussed mostly in context of its efficacy and safety profile. The other prominent topics of discussion included clinical trial studies, approval for COVID-19 treatment, ineffectiveness, awareness, accessibility and costs. HCQ mentions peaked in July 2020 after Dr Stella Immanuel, a paediatrician and religious minister supported by former President Trump, made a claim on its effectiveness in treating COVID-19. The top shared articles for HCQ highlighted the political agenda behind the use of the HCQ for COVID-19 as well as safety related articles [60,61,62]. Sentiment analysis revealed 55% of posts being positive and 45% negative. Healthcare professionals and other public users mostly voiced their opinions about the risk associated with using HCQ for COVID-19 treatment.

### 2.5. Colchicine

Colchicine was included in less than 1% of total social media posts analysed from January 2020 until June 2021, totalling 538 posts (Figure 4). There were two major peaks seen for colchicine discussions (Figure 4A(i,ii)), in July 2020 and January 2021. The prominent topic of discussion related to the medicine’s effectiveness in treating patients with COVID-19. Despite colchicine’s low overall contribution, social media mentions rose in January 2021 following a study led by the Montreal Heart Institute proving effectiveness of the drug in managing the symptoms of COVID-19 [63]. The top shared articles for colchicine covered the findings from the UK Recovery trial for colchicine administration among patients hospitalised in the UK for COVID-19 [64].

Trends or increases in social media mentions and colchicine utilisation were limited (Figure 4B, Appendix A). Given the disproportionately high baseline utilisation of colchicine, and conversely the low number of social media mentions, it is likely that any social media driven utilisation may not have driven utilisation to such a large extent.

### 2.6. Social Media Trendlines and Reporter Segmentation Analysis

In general, the increases in social media trendlines correlated well with increases in utilisation for two more popular medicines, namely, chloroquine and ivermectin. Even though the initial debate for HCQ may have been initiated by political figures including former President Trump, the highest surges were seemingly due to local politicians in South Africa (Figure 5i,ii).

When examining the sources of social media mentions, the majority (85%) of the reporters in social conversations who shared their opinion on repurposed medicines for COVID-19, the therapeutic outcomes and related side effects, were unidentifiable. Six percent of mentions were from news reporters, approximately 6% were from patients and caregivers, less than 3% were from HCPs including physicians and the remainder (<1%) were contributed by both pharmacists and other miscellaneous organisations. Figure 6 presents the overall reporter segmentation from 1070 relevant conversations. HCPs mostly answered queries and recommended management options to educate online users. 

Sentiment analysis revealed that online users expressed a more positive sentiment towards ivermectin and HCQ. No sentiments were reported for colchicine. The drivers of positive sentiments were effectiveness, their easy availability and limited cost issues. The drivers of negative sentiments were ineffectiveness, side effects, fatality, and inaccessibility. HCPs and the public’s perceived risk of HCQ were higher than its benefits, while the perceived benefit for ivermectin was higher than its risks. Ivermectin was perceived to be effective in providing quick relief to COVID-19 symptoms.

### 2.7. Clinical Trial Analysis

Only three trials involving the repurposed medicines in question were registered in South Africa. The total number of trials registered for HCQ/chloroquine, colchicine and ivermectin on clinicaltrials.gov (accessed on 1 February 2022) was 192, 30 and 60, respectively. The peak for the trendline of the chloroquines, which included chloroquine and hydroxychloroquine, was in April 2020, right at the beginning of the pandemic (Figure 7) and this does correlate with two of the three HCQ social media post peaks in March and May 2020. 

Whilst colchicine-registered trials also peaked between March and May 2020, they were far fewer than the HCQ number, and no further peaks were found. Interestingly, ivermectin had an initial peak in registration numbers between May and June 2020, with a smaller peak seen in January 2021, correlating with the January 2021 social media post peak.

Figure 8 indicates that although many trials were registered for the chloroquines, 68 (35%) of these were either withdrawn, terminated, or suspended in contrast with two (3.3%) ivermectin trials. Of those trials which were completed, results were available for 82.8%, 25% and 76.4% of the chloroquine, colchicine and ivermectin, respectively (Appendix A).

## 3. Discussion

The drive to reduce the morbidity, mortality and the spread of the virus causing COVID-19 has seen the public adoption of several repurposed medicines with limited robust evidence since the start of the pandemic. The premise for the adoption of these repurposed medicines was initially driven either by their potential effectiveness based on in vitro studies or due to previous successes with other coronaviruses [36,65,66]. However, the results from multiple robust clinical trials assessing the therapeutic potential of these re-purposed medicines have shown no clinical evidence of patient benefit to indicate their usefulness in improving the outcomes of patients treated for COVID-19 [35,41,43,45,67]. Despite this, the use of repurposed medicines such as ivermectin and HCQ are still being promoted on social media platforms, the most popular of which was ivermectin in our study, accounting for 55% of the social media mentions. This suggests that public opinion is being used to supplement and, in some cases, replace medical advice from reputable sources. The drivers underpinning this hypothesis are multi-faceted and are likely routed in a combination of public distrust of the government-backed medical regulating authorities and inaccessible or incomprehensible reliable medical information.

The COVID-19 pandemic has necessitated the implementation of stricter measures surrounding societal control and surveillance [68,69,70]. The implementation of these measures is correlated with increasing distrust in the authorities responsible for their enactment, a phenomenon which has been documented before [71]. Alongside this, information pertaining to developments at the forefront of reputable medical-scientific research still remains predominantly within scientific journals, which are often shrouded in complicated terminology or locked behind paywalls. As a result, this vital information is largely inaccessible to the average member of society who, instead, receives updates from simplified, and at times, over-hyped echoes in the media. These media reports may incorrectly portray preliminary developments in a study regarding repurposed medicines as being validated, thereby influencing public opinion, which in turn is amplified by reiterating mentions on social media. This needs to be addressed and recognised going forward, involving all key stakeholder groups including governments and HCPs especially as the majority of the responders (85%) were unidentifiable. Community pharmacists among others can play a key role here in terms of providing robust advice as they are often the first contact point for minor and other illnesses, which, in the current pandemic, included advice regarding hygiene measures as well as advice on various antimicrobials [72,73,74,75]. This is important given the extent of misinformation from social media and other platforms.

Social media analysis, while useful in evaluating public sentiment towards certain topics, still carries certain limitations. For instance, disparity in contributing populations may skew data and misrepresent true public opinion [76]. This has been shown before in a study on social media influence pertaining to vaccine opinion where those who were more often exposed to negative opinions were more likely to subsequently post negative opinions [77,78]. Similarly, this phenomenon may occur within the current pandemic as distrust in government vaccination campaigns may push people towards adoption of alternative medical countermeasures. This also needs to be addressed going forward if countries are to truly benefit from developments of potential preventative medicines and treatments, including vaccinations. We have seen that misinformation regarding the pandemic increases morbidity, mortality and costs, including the mental health of the population, with concerns likely to continue if current vaccine hesitancy rates remain [17,18,39]. This needs to be addressed with pro-active approaches.

However, understanding the influence of social media and other factors on the public perception and utilisation of repurposed medicines is complicated. Even when using a mixed methods approach, barriers in the accessibility of information pertaining to both social media mentions and utilisation are significant. Within African countries, widespread community access to social media and internet infrastructure is still limited [79,80]. However, this is changing [7]. We will continue to monitor this situation as well as make sure guidelines for treating patients particularly in hospitals in the current and future pandemic are up-to-date and enacted upon through undertaking utilisation research. This is ongoing in Africa, building on research efforts in other low and low–middle income countries [81,82]. In addition, we will continue to research and monitor utilisation patterns for re-purposed medicines among community pharmacies across South Africa and wider, building on previous research [11,30,31,74,83]. Such activities should further guide policy in this complex area including encouraging more teaching and emphasis on evidence-based medicine approaches in the curricula of healthcare universities and post qualification [84].

We are aware of a number of limitations with this study. The first limitation is that only selected social media platforms were used, namely, Twitter, Facebook, Instagram, News platforms, YouTube, and Blogs. As a result, generalisations cannot be made. It was also difficult to find a starting point to build on as most of the studies conducted regarding the impact of social media were performed as part of social science studies and typically did not include data on the utilisation of repurposed medicines. A third limitation was the necessary short period of time of the study given the time scales involved and the need to document our findings to provide future guidance. As a result, the generalisation of a relationship between the use of social media on the utilisation of repurposed drugs needs to be treated with caution until more data becomes available. Fourthly, there is an absence of true patient-level consumption data. However, drug utilisation data collected for a developing country such as South Africa using IMS Health data and other sources could potentially be regarded as a reliable source of information when surveillance networks are either missing or weak. Lastly, we are aware we only conducted this study in South Africa. Despite these limitations, we believe our findings are robust and provide direction to all key stakeholder groups going forward in South Africa and wider.

## 4. Materials and Methods

### 4.1. Study Design and Setting

This study was conducted within the geographical region of South Africa and followed a retrospective quantitative and qualitative approach. Three main data sources were used to investigate the potential effect that social media may have had in terms of influencing the use and research of HCQ, colchicine and ivermectin during the COVID-19 pandemic (Figure 9). 

Firstly, social media listening tools provided by IQVIA were used to investigate patterns of social media engagement by the South African public regarding the three medicines (from January 2020 to June 2021). 

Past conversations and social media posts were reviewed to evaluate the general public’s understanding and perceptions of repurposed medicines to prevent, manage and treat COVID-19 with regard to access, product use and beliefs. The digital channels analysed included Twitter, Facebook, Instagram, News platforms, YouTube, Blogs and Forums and all languages were considered. See Figure 9 for an overview of the process. 

Secondly, a retrospective review of the utilisation patterns of the three repurposed medicines, ivermectin, colchicine, and HCQ, in South Africa for the period prior to the COVID-19 pandemic (May 2018 to December 2019) and the period during the COVID-19 pandemic (January 2020 to June 2021) was undertaken. 

Lastly, clinical trial data was obtained from the United States National Library of Medicine clinical trials website (clinicaltrials.gov, accessed on 1 February 2022) and the South African clinical trials registry website (sanctr.samrc.ac.za) to determine whether an increase in social media posts led to an increase in the number of clinical trials registered for these three medicines in question. In addition, the clinical trials themselves may have generated an increase in social media discussions concerning these re-purposed medicines. 

### 4.2. Data Collection 

#### 4.2.1. Social Media and Utilisation Data

Retrospective data was collected from January 2020 until June 2021 from publicly available social media posts. Social media platforms were analysed by IQVIA using keywords and hashtags (#). These keywords included ‘hydroxychloroquine’, ‘colchicine’, ‘ivermectin’ and ‘covid19′. IQVIA’s social media listening tools recorded all mentions of these medicines during the COVID-19 pandemic. Social media posts mentioning these medicines were collated temporally over the duration of the pandemic for investigation alongside drug utilisation patterns. By using separate keywords for each medicine, separate timelines were formed to determine any patterns among the public regarding social media influence.

A report was subsequently obtained from IQVIA including all the relevant data. This software platform tracks for example, brands, categories and sentiments. In these reports, social media posts were separated according to demographics, channels, medicines, and the professions of users.

IQVIA provided data on all the specified medicines, sourced from three different platforms. Although the database does not report data on standard units of consumption, i.e., daily defined dose (DDD), it shows information including the sales, de-identified prescription data, medical claims, and electronic medical records. The three different platforms are: The Wholesaler Sell-In Data For Private Market In South Africa (Platform I), Prescription And Over-The-Counter Use Private Market Dispensed Sell-Out Data (Platform II), and The Public State Sector Sell-In Data (Platform III).

#### 4.2.2. Clinical Trials

Retrospective data on clinical trials registered for the three medicines in question were obtained from clinicaltrials.gov (accessed on 1 February 2022) and sanctr.samrc.ac.za. Key words used in the searches included the named medicines and COVID-19 or SARS-CoV-2. “Hydroxychloroquine” as well as “chloroquine” were used to capture all pertinent trials for HCQ. Trials recruiting patients from 1 January 2020 to December 2021 were included. All available information such as the trial nature, number of enrolled participants (projected or completed), registration date, phase, participating institutions and results if any (including links to published manuscripts), as well as whether the study was suspended, withdrawn, or terminated were recorded.

### 4.3. Processing of Data and Statistics

#### 4.3.1. Social Media

All data was filtered, auto-categorised and duplicates removed to establish a unique data set. Trendlines of the social conversation of each repurposed medicine were formed illustrating the number of times each medicine was discussed or mentioned. The data collected was separated into different themes (thematic analysis) of discussion and their sentiment analysed. Perceptions regarding the three repurposed medicines were also highlighted in the report. The research team used sentiment analysis to allow reflectivity of the research to investigate perceptions and affirmations. Sentiment analysis was conducted by extracting data from the tool based on specific therapy and brand related keywords.

The following hashtags (#s) were used: #ivermectin, #covid19, #hydroxychloroquine, #coronavirus, #sabcnews, #covid, #southafrica, #covid_19, #ivermectin4sa and #ivermectinworks. ‘#ivermectin’ and ‘#covid-19′ were to research social media use.

These data were subsequently filtered and contextualized to obtain relevant posts specific to the study. Each post was manually analysed by life science experts to obtain actionable insights. Subsequently, the proportion of positive vs. negative sentiment associated with the brands in scope that is expressed by consumers was analysed. Following this, the main drivers or reasons for positive/negative online sentiment related to the brands were analysed. This analysis was based on perceptions and affirmations. For example, phrases such as “very effective”, “was better in 3 days”, and “condition improved” were perceived as an expression of positive sentiment towards the brand in scope, whereas phrases such as “do not work”, “does not help”, “is so unsafe” were perceived as an expression of negative sentiment.

For the utilisation data, the study variables were as follows: active pharmaceutical ingredients of the medicine were classified using the WHO ATC Classification system, and the NAPPI (National Approved Product Pricing Index) codes; Treatment duration; and Prevalence.

Utilisation data are presented in moving annual total (MAT) units over-the-counter (OTC) Rand value; the MAT units are the cumulative sum of a variable, such as sales or units of a specific product, over the course of the previous 12 months. MAT units are a rolling yearly sum that change at the end of each month. The results were also presented as the class unit market share, a measure of the units sold as a percentage of total sales. It is calculated by dividing the units sold over a specific period by the total sales in the very same period. Lastly, the results were presented in compound annual growth rate (CAGR), which is calculated using the following formula: $\mathrm{CAGR}\;={\left(\frac{SUEnd}{SUStart}\right)}^{\left(\frac{1}{N}\right)}–\;1$, with SUEnd representing the number of standard units for the last reported year, SUStart depicting the total number of standard units for the first reported year, and N is the number of years between the first and the last reported year [85].

#### 4.3.2. Clinical Trials

Once all the data was collated in Microsoft Excel™ spreadsheets, tallies of the number of international trials registered on a monthly basis for each drug, from 1 January 2020 to June 2021 was displayed in graph format. The number of trials still active and recruiting, and the number terminated or withdrawn for each drug, was also graphically portrayed.

#### 4.3.3. Triangulation and Trustworthiness of the Data 

We considered the criteria outlined by Lincoln and Guba (1985) to ensure the trustworthiness of the data [86]. Understanding the influence of social media and other factors on the public’s perception and utilisation of repurposed medicines is complicated. To adequately address this objective, a mixed methods approach was used, which integrates a variety of data types from multiple sources, and more accurately triangulate underlying drivers thereof, including both qualitative and quantitative research methods. This approach circumvents some of the limitations of single-method designs. For instance, market indicators as active pharmaceutical ingredients for each repurposed medicine were classified using the WHO ATC Classification system, and the National Approved Product Pricing Index (NAPPI) codes. These classifications were used to group each medicine according to formulation so that utilisation of these repurposed medicines may be tracked across different manufacturers.

The quantitative market indicators were monitored in combination with qualitative data from social media listening tools, collected as described above, to identify the social mechanisms underpinning utilisation. Conversation on public forums and social media posts were reviewed to evaluate the general public’s understanding and sentiment towards repurposed medicines to prevent manage and/or treat COVID-19 with regard to access, product use and personal beliefs. The digital channels analysed included Twitter, Facebook, Instagram, News, YouTube, Blogs and Forums (refer to Table 1) with consideration for differing languages. Data from social media listening tools were further combined and compared with data pertaining to clinical trial initiation.

Furthermore, to ensure trustworthiness of the thematic analysis, distinct steps were followed. These steps include familiarization of the data, generating codes, and thematic analysis; subsequently, reviewing the themes (this included the sentiment analysis), defining, and naming the themes (the sentiments were classified as either positive or negative) and finally producing the report (this paper). These steps were interrelated and occurred simultaneously throughout the research process.

By combining data from several sources, the influence of social media on the overall patterns of repurposed drug utilisation could be triangulated, to ensure the credibility and conformability of the data. As is outlined by the preceding sections and to ensure trustworthiness of the thematic analysis we included the trustworthiness criteria outlined by Lincoln and Guba (1985). The trustworthiness criteria include six different and distinct steps. The steps include familiarization of the data, generating codes, thematic analysis, then reviewing the themes (this included the sentiment analysis), defining, and naming the themes (the sentiments were classified as either positive or negative) and finally producing the report (this paper). We recognize that these steps were interrelated and occurred simultaneously throughout the research process [86].

## 5. Conclusions

We believe this study provides unique insight into the impact of social media as a possible driver of utilisation of medicines currently not registered for the management of COVID-19. Ivermectin was the most important of the three repurposed medicines in our study. HCQ mostly dominated in the early part of the pandemic, whereas colchicine showed a different trendline influenced by a clinical trial that showed some positive results. However, ivermectin was the most interesting of the three medicines under investigation, with the social conversation trendline picking up in 2021.

The struggle to get ivermectin approved in South Africa and wider, despite limited current evidence, continues to be a conversation item in social media. Overall, this study highlights the important role that HCPs and scientists play in educating the public to enhance the rational use of medicines based on available data in the midst of a pandemic. Otherwise, there could be an increase in morbidity, mortality and costs as seen with HCQ across countries. Furthermore, it is important to continue monitoring these trends as the COVID-19 vaccination rate increases and the conversations of how the pandemic may end, begin to emerge. It will also be important to monitor if other unregistered medicines enter the social media conversations and for Governments and HCPs to become more pro-active in dealing with this. These are important considerations for the future.

## Figures and Tables

**Figure 1 antibiotics-11-00445-f001:**
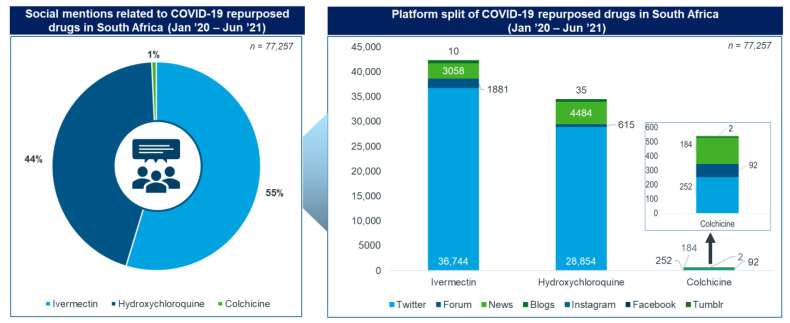
Social mentions of repurposed medicines.

**Figure 2 antibiotics-11-00445-f002:**
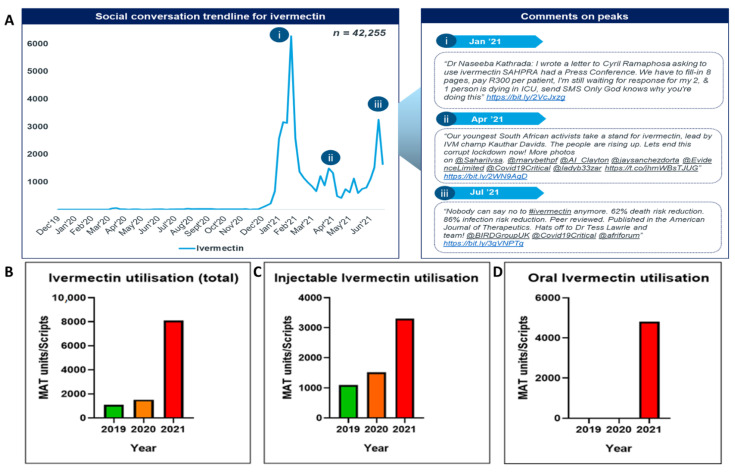
Social conversation trendline vs. moving annual total (MAT) units/scripts for ivermectin. (**A**) Social conversation trendline for Ivermectin and comments on peaks. (**B**) Ivermectin utilisation (total). (**C**) Injectable Ivermectin utilisation. (**D**) Oral Ivermectin utilisation.

**Figure 3 antibiotics-11-00445-f003:**
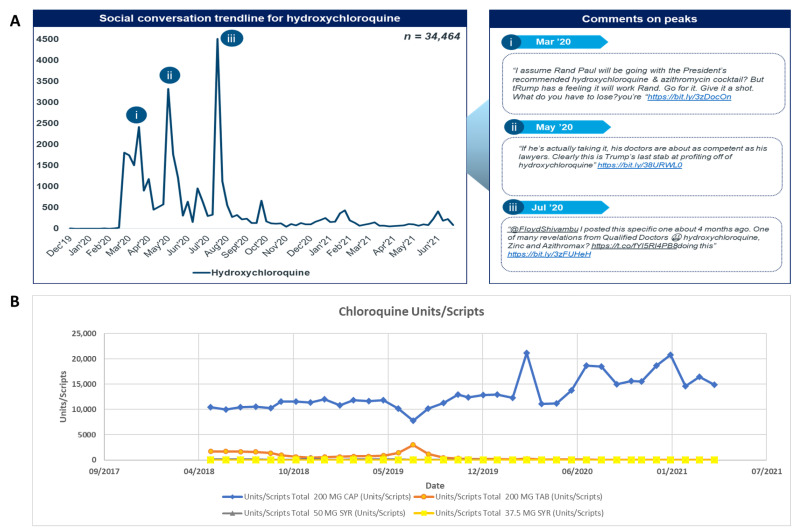
(**A**) Social conversation trendline (**B**) utilisation trendline as units/scripts for chloroquine.

**Figure 4 antibiotics-11-00445-f004:**
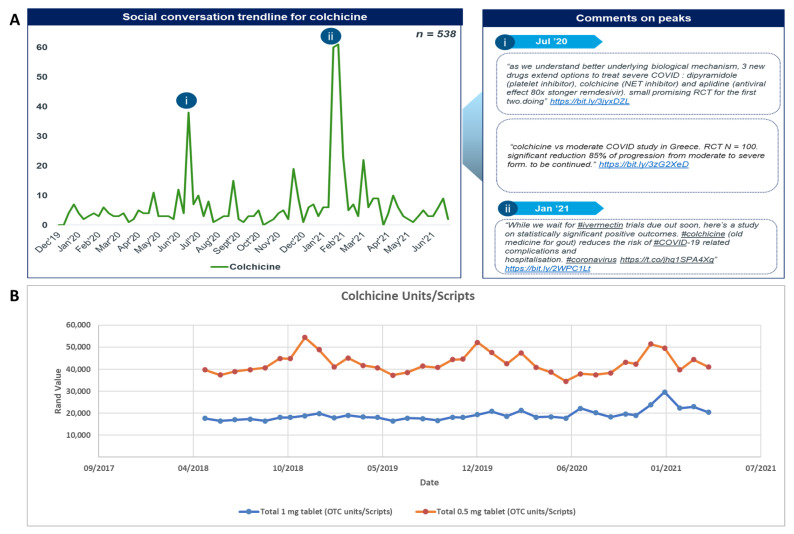
(**A**) Social conversation trendline for colchicine and (**B**) utilisation trendline as units/scripts for colchicine.

**Figure 5 antibiotics-11-00445-f005:**
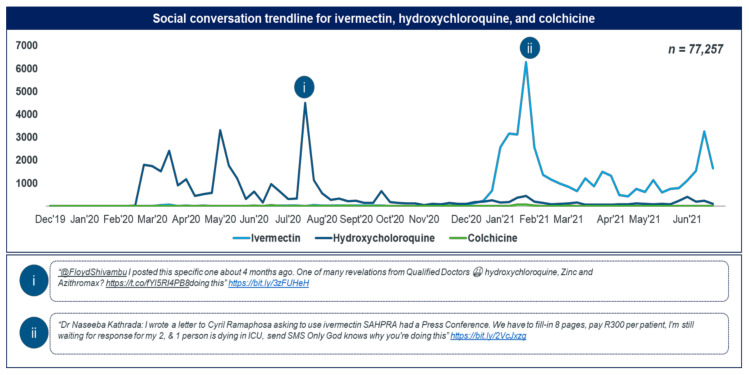
Combined social conversation trendline for ivermectin, HCQ, and colchicine.

**Figure 6 antibiotics-11-00445-f006:**
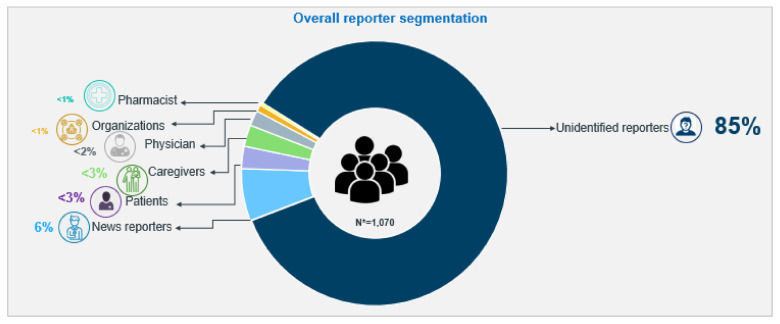
Overall reporter segmentation.

**Figure 7 antibiotics-11-00445-f007:**
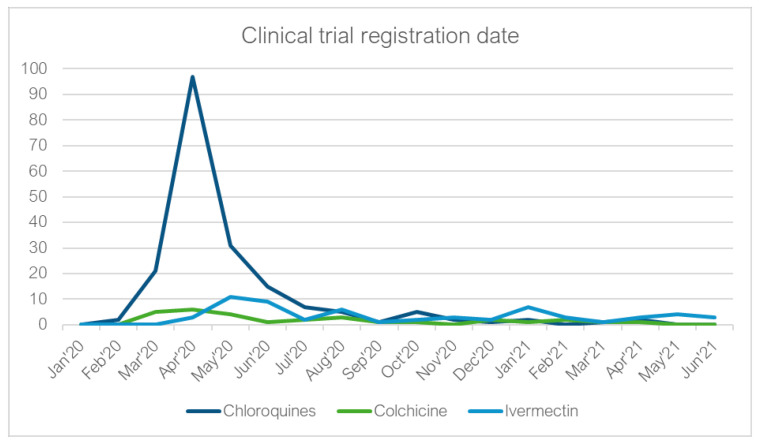
The number of clinical trials registered each month from January 2020–June 2021 for the chloroquine, colchicine and ivermectin.

**Figure 8 antibiotics-11-00445-f008:**
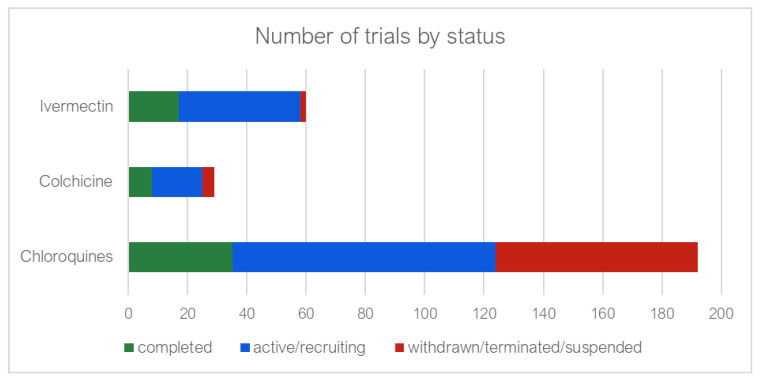
Number of clinical trials completed, still active and either withdrawn, suspended or terminated.

**Figure 9 antibiotics-11-00445-f009:**
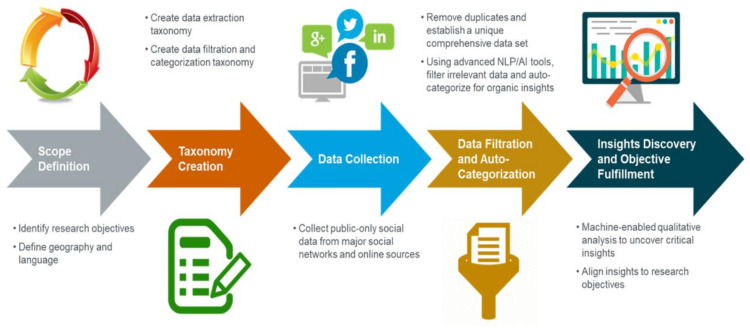
Social media listening, an overview of the methodology used.

**Table 1 antibiotics-11-00445-t001:** Summary of posts analysed.

Stratification	Total	Twitter	News	Forum	Blogs	Tumblr
Total number of posts collected	77,257	65,850	7726	2588	1046	47
Number of posts filtered and contextualized to authors/reporters	25,021	15,460	6540	2150	824	47
Total number of posts analysed	11,688	5800	5076	560	205	47
Number of posts identified to be relevant	6884	1558	5076	44	205	1
Number of unique authors/reporters identified from relevant posts	1070	907	70	28	64	1

## Data Availability

Additional data is available in the Appendix A. Any further data is available on reasonable request from the corresponding authors.

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
