# Peer review of "Social Media and COVID-19—Perceptions and Public Deceptions of Ivermectin, Colchicine and Hydroxychloroquine: Lessons for Future Pandemics"

_antibiotics, 2022, doi:10.3390/antibiotics11040445_

Round 1

Reviewer 1 Report

This is an innovative and constructive way to view the influence of social media on inappropriate prescribing during COVID-19. While this paper is overall of high quality, I have a few suggestions that might be helpful. Overall, this paper is quite lengthy and could improve by cutting down and focusing the content.

Introduction

  • This whole section is very long and takes on the quality of a review article, rather than an introduction to a scientific study. Consider removing section 1.1 and focus on 1.2 as the introduction. I would remove the paragraph about antibiotics, as this is a deterrent and misleading from the point of the article, which is to focus on HCQ, ivermectin and colchicine. In its place, you can consider having condensed information about the misuse of these three drugs (pulled from section 1.1) instead. That should help set-up the last paragraph of this section nicely.

Methods

  • Would highly encourage the use of the SRQR statement on what information to include in the methods section (and how to conduct) a qualitative analysis of data.
  • Can remove lines 471-475 and 501-510, as I feel that this information is outside the scope of the study as outlined by the study’s purpose.
  • Please clarify what you mean by “business objectives”.

Results

  • Love figure 1. The colchicine stacked bar graph is hard to understand, as even with the numbers, it’s still unclear which number corresponds to each social media platform. Would consider creating a broken y axis to allow colchicine bar to expand to see better. Shouldn’t be a problem with how prevalent HCQ and ivermectin are.
  • Also love figure 2. Visual representations are extremely helpful. Thank you.
  • If providing correlation, please provide the proper statistics to support this information. The methods indicate statistics were performed, however, these data were not presented in the results. Please provide this data.
  • Please spell out MAT (line 261) and CAGR (line 268) prior to abbreviating (full names appear in the methods section, which is after the discussion).
  • The results section should be straight data, not conjectures (lines 275-277).
  • I’m not sure why section 2.7 was included, as this is beyond the scope of the intent of this article (social media influence on HCQ, ivermectin, colchicine use). Please remove this section, as well as figure 7 and 8.

Author Response

Reviewer 1:

Comments and Suggestions for Authors

This is an innovative and constructive way to view the influence of social media on inappropriate prescribing during COVID-19. While this paper is overall of high quality, I have a few suggestions that might be helpful. Overall, this paper is quite lengthy and could improve by cutting down and focusing the content.

Author comments: Thank you for your comments and suggestions. We hope we have satisfactorily addressed these.

  1. A) Introduction - This whole section is very long and takes on the quality of a review article, rather than an introduction to a scientific study. Consider removing section 1.1 and focus on 1.2 as the introduction. I would remove the paragraph about antibiotics, as this is a deterrent and misleading from the point of the article, which is to focus on HCQ, ivermectin and colchicine. In its place, you can consider having condensed information about the misuse of these three drugs (pulled from section 1.1) instead. That should help set-up the last paragraph of this section nicely.

Author comments: Thank you for your comments and suggestions. As seen, we have now started with old 1.2. However, we have included some parts of old 1.1. as we believe this helps set the scene/ rationale for the study – especially given limited evidence for the effectiveness of these 3 re-purposed medicines along with concerns. We hope you agree.

  1. B) Methods

  1. a) Would highly encourage the use of the SRQR statement on what information to include in the methods section (and how to conduct) a qualitative analysis of data.

Thank you for this comment. Below is how we have addressed this, which has resulted in changes to the Methodology. We hope these issues and changes are acceptable.

Qualitative approach and research approach

We attempted to address this with the section headed Triangulation of Data:

Understanding the influence of social media and other factors on the public’s perception and utilisation of repurposed medicines is complicated. To adequately address this objective, a mixed methods approach which integrates a variety of data types from multiple sources, is appropriate in more accurately triangulating underlying drivers thereof. Both qualitative and quantitative research methods were implemented in the study. This approach circumvents some of the limitations of single-method designs. For instance, market indicators as active pharmaceutical ingredients for each repurposed medicine were classified using the WHO ATC Classification system, and the National Approved Product Pricing Index (NAPPI) codes. These classifications were used to group each medicine according to formulation so that utilisation of these repurposed medicines may be tracked across different manufacturers.

These quantitative market indicators were monitored in combination with qualitative data from social media listening tools, collected as described above to identify the social mechanisms underpinning utilisation. Conversation on public forums and social media posts were reviewed to evaluate the general public’s understanding and sentiment towards repurposed medicines to prevent manage and/or treat COVID-19 with regards to access, product use and personal beliefs. The digital channels analysed included Twitter, Facebook, Instagram, News, YouTube, Blogs and Forums (refer to Table 1) with consideration for differing languages.

Data from social media listening tools were further combined and correlated with data pertaining to clinical trial initiation. By combining data from several sources, the influence of social media on the overall patterns of repurposed drug utilisation could be triangulated.

Research characteristics and reflexivity

We hoped that we have already addressed this under the section titled: “Processing of Data “with the following. We added a sentence that may include more reflectivity.

All data was filtered, auto-categorised and duplicates removed to establish a unique data set. Trendlines of the social conversation of each repurposed medicine were formed illustrating the number of times each medicine was discussed or mentioned. The data collected was separated into different themes of discussion and their sentiment analysed. Perceptions regarding the three repurposed medicines were also highlighted in the report.

The research team used sentiment analysis to allow reflectivity of the research to investigate perceptions and affirmations. Sentiment analysis was conducted by extracting data from the tool based on specific therapy and brand related keywords. 

The following hashtags (#’s) were used #ivermectin, #covid19, #hydroxychloroquine, #coronavirus, #sabcnews, #covid, #southafrica, #covid_19, #ivermectin4sa and #ivermectinworks. ‘#ivermectin’ and ‘#covid-19’ to research social media use.

This data was subsequently filtered and contextualized to obtain relevant posts specific to the business objectives. Each post was manually analysed by life science experts to obtain actionable insights. Subsequently, the proportion of positive vs negative sentiment associated with the brands in scope that is expressed by consumers was analysed. Following this, the main drivers or reasons for positive/ negative online sentiment related to the brands were analysed. This analysis was based on perceptions and affirmations. For example: phrases such as “very effective”, “was better in 3 days”, “condition improved” were perceived as an expression of positive sentiment towards the brand in scope.  Whereas, phrases such as “do not work”, “does not help”, “is so unsafe” were perceived as an expression of negative sentiment.

Context

We hoped that this was addressed in study design and setting – for the different aspects of the study.

Sampling Strategy

This was not used in the study and no sampling was needed. Our approach is described in the section “processing of data”

Ethical Issues

Included under the heading:

Institutional Review Board Statement: This study was approved by the ethics committee at the University of Pretoria (ethics approval number: 252/2021; ethics application number: 251/2021).

Due to change in privacy policies on Instagram, only publicly available posts from business accounts and hashtags could be tracked, while on Facebook only posts from public pages can be analysed. No personal data from Facebook, Instagram or YouTube were recorded and could be identified from the analysis.

Informed Consent Statement: Not applicable as no patients were involved in this study; consequently, informed consent was not sought. This is in line with previous studies conducted by the authors in related areas [7,12,12,26,58,87,115,116] However, ethical approval was obtained for the study

Data Availability Statement: Additional data is available in the Supplementary Material. Any further data is available on reasonable request from the corresponding authors

Data collection methods

Included under the heading 4.2

Data collection instruments and technologies

Same as above

Units of study

All units are included and introduced in the methods section

Data processing

Included under section 4.3

Data analysis

Included under section 4.3

Techniques to enhance trustworthiness

The following was included under the heading triangulation and trustworthiness.

As is outlined by the preceding sections and to ensure trustworthiness of the thematic analysis we included the trustworthiness criteria outlined by Lincoln and Guba (1985). The trustworthiness criteria include six different and distinct steps. The steps include familiarization of the data, generating codes, thematic analysis, then reviewing the themes (this included the sentiment analysis), defining, and naming the themes (the sentiments were classified as either positive or negative) and finally producing the report (this paper). We recognize that these steps were interrelated and occurred simultaneously throughout the research process.  

  1. b) Can remove lines 471-475 and 501-510, as I feel that this information is outside the scope of the study as outlined by the study’s purpose.

Author comments: Thank you, agreed and removed 471-475 and these lines 511-513 “IQVIA, formerly Quintiles and IMS Health, Inc., is an American multinational company serving the combined industries of health information technology and clinical research. “The rest we kept for contextualization

  1. c) Please clarify what you mean by “business objectives”.

Author comments: Thank you - we have removed the phrase as it bears no relevance

  1. C) Results

Love figure 1. The colchicine stacked bar graph is hard to understand, as even with the numbers, it’s still unclear which number corresponds to each social media platform. Would consider creating a broken y axis to allow colchicine bar to expand to see better. Shouldn’t be a problem with how prevalent HCQ and ivermectin are.

  1. a) Thank you for this positive comment – appreciated. We have now expanded the colchicine bar and hope this is now acceptable.

  1. b) Also love figure 2. Visual representations are extremely helpful. Thank you.

  1. c) If providing correlation, please provide the proper statistics to support this information. The methods indicate statistics were performed; however, these data were not presented in the results. Please provide this data.

Author comments: Thank you for the comment, we steered clear of the word “correlations” and only refer to ‘increases’ or following a trend in view of the difficulties in showing true associations. Consequently, we refrained from the terminology. We hope this is acceptable.

  1. d) Please spell out MAT (line 261) and CAGR (line 268) prior to abbreviating (full names appear in the methods section, which is after the discussion).

Author comments: Thank you, addressed as requested

  1. e) The results section should be straight data, not conjectures (lines 275-277).

Author comments: Thank you, lines removed as requested

  1. f) I’m not sure why section 2.7 was included, as this is beyond the scope of the intent of this article (social media influence on HCQ, ivermectin, colchicine use). Please remove this section, as well as figure 7 and 8

Author comments: Thank you for this comment. However, we beg to differ if we can as one impact of the social media campaigns could be increasing pressure on healthcare professionals and governments to conduct clinical trials for these 3 repurposed medicines. In addition – their very nature may have increased social media traffic regarding these re-purposed medicines. We hope you now agree with this and can keep this section and comments in the paper.

Reviewer 2 Report

The bumpy ride from social perceptions to public deceptions for unregistered medicines for the management of COVID-19 and the implications

Schellack et al.

This is not the “usual type” of science papers that I have reviewed for a scientific journal.  Whether such a paper is appropriate for the journal, “Antibiotics” is to be decided by the editors. However, I found it to be a timely paper on the spread of misinformation on social media.  The authors have done an appropriate analysis of the social media data on the topic.  The paper has been presented well although at times I found it to be too long. 

Here are some minor comments on the manuscripts:

Page 1 line 29 and Page 3 lines 105, 106: “concerns with”. Use of the word “concerns” this way does not appear familiar to me.  Please check the correct preposition (with, by, about, regarding etc.,) to use when concern is used as a noun.

Page 1 line 37: Add a comma after “Similarly”

Page 1 line 37-38: “Similarly….politicians” This sentence is incomplete.  It does not have a verb.  Suggestion: change, “with social media interest enhanced” to “, social media interest was enhanced”

Page 1 line 38: “Sentiment analysis…..effectiveness” Meaning of this sentence is not clear.

Page 1 line 39: “however….mentions” This is missing a verb.

Page 1 line 40: Change “is the” to “is that the”

Page 2 line 78 and 82: “certainly initially” should be preceded and followed by comma.

Page 2 line 83: Change “patent” to “patient”

Page 2 line 97: “similarly….regimens”  Please differentiate between drugs that were approved by some reliable regulatory agencies such as the FDA and those that were merely promoted in social media.  Also comment on why drugs that went through the rigorous process of approval did not perform well.  Do some of these, such as remdesivir have some effectiveness?

Page 3 line 112-113: “This is…viral replication.” Unnecessary repetitive use of words.

Page 3 line 113: Change “Despite the limitations of current studies” to “Despite the reports in several current studies”

Page 3 line 138: Change “Having said this society” to “The society”

Page 3 line 145: Change “failing” to “which failed”

Page 3 line 148: Change “spreading this” to “spreading it”

Page 3 line 148: “Such activities are likely to become more widespread to reduce fake news” This sentence is not clear.  Which activities? The spreading or making it illegal?  The use of “to reduce fake news” is also not clear.  Please modify the sentence.

Page 4 line 164: Change “public such as the pandemic prompt” to “public, such as the pandemic, prompt”

Page 4 line 199-201: “This….antibiotics.” The sentence is incomplete and grammatically incorrect. Suggestion: “This is especially important, given the potential for increasing morbidity, mortality and costs with inappropriate use of repurposed medicines to manage patients with COVID-19, and will also mitigate against frequent inappropriate prescribing of antibiotics.

Page 6 Fig. 1: There are too many colors that are not distinguishable from each other especially when the numbers are low.  For example for colchicine, it is not clear what the five numbers represent.  I think presenting this in a tabular form will be more clear and informative. 

Page 7 three places: “MAT” The abbreviation has not been introduced/explained.

Page 7 line 268: “CAGR” First mentioned in Page 7 but explained in Page 17.  It will be good to write the full form in page 7 and mention that it is explained later in Sect. ??)

Page 8 line 274: Change “increased” to “increases”

Page 8 line 276: “as he, on multiple occasions regarding HCQ”.  Incomplete sentence.  Missing a verb.

Page 10 line 331: Change “6% were identified as patients” to “6% were from patients”

Page 10 line 332: Change “3% were healthcare” to “3% were from healthcare”

Page 10 line 332: “and the remainder organisations (<1%)” Not clear what this means.  The authors probably mean “and the remainder (<1%) were other miscellaneous organisations”

Page 10 line 332: “remainder”. There is no remainder because the total is already more than 100 % (85+7+6+3 = 101%). The numbers don’t match with those in Fig. 6 because patients, caregivers and healthcare professionals have been combined.

Page 13 line 380: Change “has shown” to “have shown”

Page 13 line 395: “pay walls”  “Paywall” is usually a single word.  I am not sure if it can be two words; please check

Page 14 line 431: “LMIC” The full form has not been mentioned anywhere.  Since this is the only time it has been used in the manuscript, the full form should be mentioned here.

Page 18 line 592: Change “end begin” to “end, begin”

Author Response

Reviewer 2

The bumpy ride from social perceptions to public deceptions for unregistered medicines for the management of COVID-19 and the implications Schellack et al.

This is not the “usual type” of science papers that I have reviewed for a scientific journal.  Whether such a paper is appropriate for the journal, “Antibiotics” is to be decided by the editors. However, I found it to be a timely paper on the spread of misinformation on social media.  The authors have done an appropriate analysis of the social media data on the topic.  The paper has been presented well although at times I found it to be too long. 

Author comments: Thank you for this positive comment and your help – appreciated. Hopefully our amendments will make it easier to read! This includes cutting down on the introduction and focusing initially on the influence of social media. In addition – principally concentrating on social media.

Here are some minor comments on the manuscripts:

  1. a) Page 1 line 29 and Page 3 lines 105, 106: “concerns with”. Use of the word “concerns” this way does not appear familiar to me.  Please check the correct preposition (with, by, about, regarding etc.,) to use when concern is used as a noun.

Author comments: Thank you, changes have been made to sentence structure to help with the clarity – hope now OK.

  1. b) Page 1 line 37: Add a comma after “Similarly”

Author comment: Thank you, addressed as requested

  1. c) Page 1 line 37-38: “Similarly….politicians” This sentence is incomplete.  It does not have a verb.  Suggestion: change, “with social media interest enhanced” to “, social media interest was enhanced”

Author comment: Thank you - comment now addressed.

  1. d) Page 1 line 37: Add a comma after “Similarly”

Author comment: Thank you, addressed as requested

  1. e) Page 1 line 38: “Sentiment analysis…..effectiveness” Meaning of this sentence is not clear.

Author comment: Thank you, clarified as requested

  1. f) Page 1 line 39: “however….mentions” This is missing a verb.

Author comment: Thank you, addressed as requested

  1. g) Page 1 line 40: Change “is the” to “is that the”

Author comment: Thank you, addressed as requested

  1. h) Page 2 line 78 and 82: “certainly initially” should be preceded and followed by comma.

Author comment: Thank you, reworded to better reflect intention

  1. i) Page 2 line 83: Change “patent” to “patient”

Author comment: Thank you, addressed as requested

  1. j) Page 2 line 97: “similarly….regimens”  Please differentiate between drugs that were approved by some reliable regulatory agencies such as the FDA and those that were merely promoted in social media.  Also comment on why drugs that went through the rigorous process of approval did not perform well.  Do some of these, such as remdesivir have some effectiveness?

Author comment: Thank you, the introduction has been, reworded and the registration processes are described. We have seen across countries that initial re-purposed medicines apart from dexamethasone have shown to have limited clinical benefit when included in robust clinical trials. Consequently – their use more driven by hype, etc., than any scientific evidence – emphasizing the desperation of citizens across countries to try and seek effective treatments when faced with considerable surges in morbidity and mortality across countries. We hope this is now acceptable.

  1. k) Page 3 line 112-113: “This is…viral replication.” Unnecessary repetitive use of words.

Author comment: Thank you, addressed as requested

  1. l) Page 3 line 113: Change “Despite the limitations of current studies” to “Despite the reports in several current studies”

Author comment: Thank you, addressed as requested

  1. m) Page 3 line 138: Change “Having said this society” to “The society”

Author comment: Thank you, addressed as requested

  1. n) Page 3 line 145: Change “failing” to “which failed”

Author comment: Thank you, addressed as requested

  1. o) Page 3 line 148: Change “spreading this” to “spreading it”

Author comment: Thank you, addressed as requested

  1. p) Page 3 line 148: “Such activities are likely to become more widespread to reduce fake news” This sentence is not clear.  Which activities? The spreading or making it illegal?  The use of “to reduce fake news” is also not clear.  Please modify the sentence.

Thank you, addressed as requested

  1. q) Page 4 line 164: Change “public such as the pandemic prompt” to “public, such as the pandemic, prompt”

Author comment: Thank you, addressed as requested

  1. r) Page 4 line 199-201: “This…antibiotics.” The sentence is incomplete and grammatically incorrect. Suggestion: “This is especially important, given the potential for increasing morbidity, mortality and costs with inappropriate use of repurposed medicines to manage patients with COVID-19, and will also mitigate against frequent inappropriate prescribing of antibiotics.

Author comment: Thank you addressed as requested included removing comments on antibiotics.

  1. s) Page 6 Fig. 1: There are too many colors that are not distinguishable from each other especially when the numbers are low.  For example for colchicine, it is not clear what the five numbers represent.  I think presenting this in a tabular form will be more clear and informative. 

Author comment: Thank you Figure 1 has been amended we hope it is easier to read now.

Page 7 three places: “MAT” The abbreviation has not been introduced/explained.

Author comment: Thank you, addressed as requested

  1. t) Page 7 line 268: “CAGR” First mentioned in Page 7 but explained in Page 17.  It will be good to write the full form in page 7 and mention that it is explained later in Sect. ??)

Author comment: Thank you addressed as requested

  1. u) Page 8 line 274: Change “increased” to “increases”

Author comment: Thank you, addressed as requested

  1. v) Page 8 line 276: “as he, on multiple occasions regarding HCQ”.  Incomplete sentence.  Missing a verb.

Author comment: Thank you, addressed as requested

  1. w) Page 10 line 331: Change “6% were identified as patients” to “6% were from patients”

Author comment: Thank you, addressed as requested

  1. x) Page 10 line 332: Change “3% were healthcare” to “3% were from healthcare”

Author comment: Thank you, addressed as requested

  1. y) Page 10 line 332: “and the remainder organisations (<1%)” Not clear what this means.  The authors probably mean “and the remainder (<1%) were other miscellaneous organisations”

Author comment: Thank you, addressed as requested

  1. z) Page 10 line 332: “remainder”. There is no remainder because the total is already more than 100 % (85+7+6+3 = 101%). The numbers don’t match with those in Fig. 6 because patients, caregivers and healthcare professionals have been combined.

Author comment: Thank you for pointing this out. This is due to the probability data round-off. The significant digits of the different sets of data and decimal points led to this. Hope this is now OK

Either provide the exact fraction or decimal of the probability or round-off the final data sets and its significant digits. News reporters were also incorrectly assigned as 7%, this has been corrected. Hope this is now OK.

  1. ai) Page 13 line 380: Change “has shown” to “have shown”

Author comment: Thank you, addressed as requested

  1. bi) Page 13 line 395: “pay walls”  “Paywall” is usually a single word.  I am not sure if it can be two words; please check

Author comment: Thank you, addressed as requested – one word is correct

  1. ci) Page 14 line 431: “LMIC” The full form has not been mentioned anywhere.  Since this is the only time it has been used in the manuscript, the full form should be mentioned here.

Author comment: Thank you, addressed as requested

  1. di) Page 18 line 592: Change “end begin” to “end, begin”

Author comment: Thank you, addressed as requested

Reviewer 3 Report

The topic is of interest addressing a reasonably current issue in this Covid-time. In addition, the manuscript well fits the scope of this journal; it is well structured, and the data are adequately presented.

However, there are some points/issues which should be clarified, revised and improved:

1) The title is too long and barely noticeable; it should be simple, concise, clear, short and impactful, pointing out chloroquine,  ivermectin and colchicine.

2) The reviewer highlights that chloroquine, ivermectin and colchicine are not medicines but active substances from medicinal products. The manuscript should be improved. For instance, medicinal products which contain ivermectin were not approved for the treatment of COVID-19…

3) Why does the social conversation trendline for ivermectin show dominant peaks in 2021? it would make sense to discuss this result and compare it with other regions (e.g. EU)

3) Comments in the main test:

L75: …originally intended indications… originally approved indications

L100: … colchicine is a medicine registered for the treatment of gout… Colchicine is a substance used for the treatment of gout

L120: Medicinal products containing HCQ, colchicine and ivermectin are not registered …

L244: Figure 2: better consumption instead of utilisation

L258: consumption data (OECD statistics, e.g. DDD - WHO ATC Classification system)

L281: …Among the four chloroquine formulations…Medicinal product formulations (dosages?) containing chloroquine. Formulation: a combination of various chemical substances with the active drug to form a final medicinal product.

L417-L418: We have seen in this pandemic that misinformation increases morbidity, mortality and costs, including the mental health of the population, with concerns likely to continue if current vaccine hesitancy rates continue [30, 63, 73]. What kind of misinformation?

L544-L546: reference, please!

Author Response

Reviewer 3

Comments and Suggestions for Authors

The topic is of interest addressing a reasonably current issue in this Covid-time. In addition, the manuscript well fits the scope of this journal; it is well structured, and the data are adequately presented.

However, there are some points/issues which should be clarified, revised and improved.

Author comments: Thank you for these positive comments and we hope we have adequately addressed the comments you made.

1) The title is too long and barely noticeable; it should be simple, concise, clear, short and impactful, pointing out chloroquine,  ivermectin and colchicine.

Author comment: Thank you – now changed. We hope this is now acceptable.

2) The reviewer highlights that chloroquine, ivermectin and colchicine are not medicines but active substances from medicinal products. The manuscript should be improved. For instance, medicinal products which contain ivermectin were not approved for the treatment of COVID-19…

Author comment: Thank you. We have kept this terminology as these substances are mentioned by their name in the numerous scientific papers that we have quoted. In addition, named as such in social media reports. Consequently, we would like to keep this terminology. However – we have re-emphasized that none of these 3 medicines have yet been approved by the authorities in South Africa. We hope this is now OK.

3) Why does the social conversation trendline for ivermectin show dominant peaks in 2021? it would make sense to discuss this result and compare it with other regions (e.g. EU)

Author comment: Thank you! Social media and news coverage popularity for the substance within South Africa peaked within that time, as did the utilization, the cause for this surge may be due to several reasons but the evidence available to us which may support these reasons would be purely speculative. We do agree that corroborating the data collected here with data from other regions would be very useful and make the conclusions within this manuscript more powerful. However, given that the social media listening tools utilized in this study only captured data from users within South Africa drawing comparisons to other regions or extrapolating observations noted here beyond their current context would we believe be inappropriate. Even if one were to manually search for reports from, for example, European news outlets, the ability to present cases either corroborating or refuting the current narrative would be too subject to biases and “cherry picking”. The use of an unbiased third party to capture large sample data is essential for undertaking a similar comparative analysis within other regions, and unfortunately, impossible within the current scope of this manuscript. We hope to address this in the future if possible.

3) Comments in the main test:

L75: …originally intended indications… originally approved indications

Author comment: Thank you, addressed as requested

L100: … colchicine is a medicine registered for the treatment of gout… Colchicine is a substance used for the treatment of gout

Author comment: Thank you, reworded for clarification

L120: Medicinal products containing HCQ, colchicine and ivermectin are not registered …

Author comment: Thank you, Reworded for clarification

L244: Figure 2: better consumption instead of utilisation

Author comment: Thank you – we have tended to use mainly consumption as this is as well accepted as utilization in recognition of the considerable debate about these 2 terms in the literature, etc. We hope this is acceptable

L258: consumption data (OECD statistics, e.g. DDD - WHO ATC Classification system)

Author comment: Thank you – as mentioned in the methodology we used data supplied by IQVIA in their units for consistency. This is different to DDDs (which the co-authors have used extensively in their other research including cross-country comparisons). However, IQVIA units are also used across countries including cross-national studies to track utilization data (and used by the Pharma Industry as well). In this case, IQVIA units more acceptable as there are currently no designated DDDs for these 3 medicines in patients with COVID-19 – especially as not indicated for this use. We also did not use the ATC classification here as we were only talking about three medicines. We hope this is OK.

L281: …Among the four chloroquine formulations…Medicinal product formulations (dosages?) containing chloroquine. Formulation: a combination of various chemical substances with the active drug to form a final medicinal product.

Author comment: Thank you, even though the doses may be similar, the formulation (as defined above) between the similar doses will be different, i.e., the formulation (of excipients) between 200mg tablets/capsules/syrups from different manufacturers will be different. We hope this is acceptable. 

L417-L418: We have seen in this pandemic that misinformation increases morbidity, mortality and costs, including the mental health of the population, with concerns likely to continue if current vaccine hesitancy rates continue [30, 63, 73]. What kind of misinformation?

Author comment: Thank you, reworded for clarification

L544-L546: reference, please –

Author comment: Thank you – now added in.

Round 2

Reviewer 1 Report

Thank you for making the suggested changes. This manuscript looks great!

Reviewer 3 Report

The questions / issues were answered properly the questions were answered properly